# Prevalence, causes and impact of musculoskeletal impairment in Malawi: A national cluster randomized survey

**Leonard Banza Ngoie**[1,2]*, **Eva Dybvik**[3], **Geir Hallan**[1,2,3,4], **Jan-Erik Gjertsen**[1,2,4],
**Nyengo Mkandawire**[5,6], **Carlos Varela**[1,2,5], **Sven Young**[1,2,4,5,7]

**1** Department of Surgery, Kamuzu Central Hospital, Lilongwe, Malawi, **2** Department of Clinical Medicine (K1), University of Bergen, Bergen, Norway, **3** The Norwegian Arthroplasty Register, Haukeland University Hospital, Bergen, Norway, **4** Department of Orthopaedic Surgery, The Norwegian Arthroplasty Register Haukeland University Hospital, Bergen, Norway, **5** Department of Surgery, College of Medicine, University of Malawi, Blantyre, Malawi, **6** School of Medicine, Flinders University, Adelaide, Australia, **7** Department of Orthopaedic Surgery, Haukeland University Hospital, Bergen, Norway

* leonardbanza2014@gmail.com

## Abstract

### Background

There is a lack of accurate information on the prevalence and causes of musculoskeletal impairment (MSI) in low income countries. The WHO prevalence estimate does not help plan services for specific national income levels or countries. The aim of this study was to find the prevalence, impact, causes and factors associated with musculoskeletal impairment in Malawi. We wished to undertake a national cluster randomized survey of musculoskeletal impairment in Malawi, one of the UN Least Developed Countries (LDC), that involved a reliable sampling methodology with a case definition and diagnostic criteria that could clearly be related to the classification system used in the WHO International Classification of Functioning, Disability and Health (ICF)

### Methods

A sample size of 1,481 households was calculated using data from the latest national census and an expected prevalence based on similar surveys conducted in Rwanda and Cameroon. We randomly selected clusters across the whole country through probability proportional to size sampling with an urban/rural and demographic split that matched the distribution of the population. In the field, randomization of households in a cluster was based on a ground bottle spin. All household members present were screened, and all MSI cases identified were examined in more detail by medical students under supervision, using a standardized interview and examination protocol. Data collection was carried out from 1st July to 30th August 2016. Extrapolation was done based on study size compared to the population of Malawi. MSI severity was classified using the parameters for the percentage of function outlined in the WHO International Classification of Functioning (ICF). A loss of function of 5–24% was mild, 25–49% was moderate and 50–90% was severe. The Malawian version of the EQ-5D-3L questionnaire was used, and EQ-5D index scores were calculated

**Data Availability Statement:** All relevant data are within the manuscript and its Supporting Information files.

**Funding:** This study was funded by Norad through the Norhed programme. However the funders had no role in study design, data collection and analysis, decision to publish, or preparation of the manuscript.

**Competing interests:** The authors have declared that no competing interests exist.

using population values from Zimbabwe, as a population value set for Malawi is not currently available. Chi-square test was used to test categorical variables. Odds ratio (OR) was calculated with a linear regression model adjusted for age, gender, location and education.

## Results

A total of 8,801 individuals were enumerated in 1,481 households. Of the 8,548 participants that were screened and examined (response rate of 97.1%), 810 cases of MSI were diagnosed of which 18% (108) had mild, 54% (329) had moderate and 28% (167) had severe MSI as classified by ICF. There was an overall prevalence of MSI of 9.5% (CI 8.9–10.1). The prevalence of MSI increased with age, and was similar in men (9.3%) and women (9.6%). People without formal education were more likely to have MSI [13.3% (CI 11.8–14.8)] compared to those with formal education levels [8.9% (CI 8.1–9.7), p<0.001] for primary school and [5.9% (4.6–7.2), p<0.001] for secondary school. Overall, 33.2% of MSIs were due to congenital causes, 25.6% were neurological in origin, 19.2% due to acquired non-traumatic non-infective causes, 16.8% due to trauma and 5.2% due to infection. Extrapolation of these findings indicated that there are approximately one million cases of MSI in Malawi that need further treatment. MSI had a profound impact on quality of life. Analysis of disaggregated quality of life measures using EQ-5D showed clear correlation with the ICF class. A large proportion of patients with moderate and severe MSI were confined to bed, unable to wash or undress or unable to perform usual daily activities.

## Conclusion

This study has uncovered a high prevalence of MSI in Malawi and its profound impact on a large proportion of the population. These findings suggest that MSI places a considerable strain on social and financial structures in this low-income country. The Quality of Life of those with severe MSI is considerably affected. The huge burden of musculoskeletal impairment in Malawi is mostly unattended, revealing an urgent need to scale up surgical and rehabilitation services in the country.

## Introduction

Musculoskeletal disease is one of the major causes of physical disability globally, yet data regarding the magnitude of this burden in developing countries is lacking [1]. One reason for this is the absence of a universal understanding of the definition of physical disability. The difficulty in defining physical disability stems from its many anatomical, physiological and pathological presentations and causes, and its intimate relation to society and the environment [2]. There have been many attempts to reach a common understanding of disability, and the World Health Organization's (WHO) publication of the International Classification of Functioning, Disability and Health (ICF) is a major step forward. The ICF classifies impairment of body structure and function, and also includes domains that measure activity and participation in society. Musculoskeletal impairment (MSI) is according to the ICF defined as ". . .*a lack of normal structure or function, or an increase in pain or discomfort in the integument, muscles, bone or joints of the body of an individual, that has lasted at least 1 month and which limits function of the musculoskeletal system*. . ." [2].

The UN Convention on the Rights of Persons with Disabilities (UNCRPD) defines disability *as "long-term physical, mental, intellectual or sensory impairments which, in interaction with various barriers, may hinder [a person's] full and effective participation in society on an equal basis with other"* [3].

There is also a lack of accurate information on the prevalence and causes of physical disability due to the lack of surveys in low-income countries (LICs) [4, 5]. The WHO estimates that the prevalence of all types of disability on a global level is around 10% [6], but this estimate does not help plan services in specific situations or countries. Realizing the challenge, Helander developed a 'Rapid Calculation of Disability Prevalence' for less developed regions of the world and estimated that 4.8% of a population will need some rehabilitation service [7].

Musculoskeletal disease encompasses a wide range of conditions resulting from various etiologies such as traumatic, infectious, inflammatory, metabolic, congenital, developmental and degenerative condition; many of which benefit from surgical interventions. Musculoskeletal disease is an important cause of morbidity and mortality, especially in LICs [8], affecting a large portion of the world's population in one form or another, with non-traumatic musculoskeletal disease estimated to account for 6.8% of all Disability-Adjusted Life Years (DALY) lost [1]. Most road injuries are musculoskeletal in nature [9], and several studies have shown the heavy burden of musculoskeletal injuries in Low and Middle Income Countries (LMICs) [10, 11]. For each person who dies from trauma, three to eight more are permanently disabled [12, 13]. Estimates from a nationwide survey in Rwanda suggested a prevalence of musculoskeletal impairment of 5.2% [8]. A similar study in Fundong district, North-West Cameroon found a prevalence of 11.2% [14, 15]. Data on the prevalence of MSI in Malawi is scarce.

There have been several surveys of physical disability in Malawi in the past [16, 17]. However, these studies have targeted small cohorts of the population and focused on disability in general, which may have led to an underestimation of the burden of MSI in the community in general. None of these previous studies has evaluated the quality of life among people with MSI. Therefore, it is imperative to use a survey methodology to estimate the prevalence of MSI in Malawi that can be extrapolated on a national level and compared to data from other countries. This data is needed for informing policy development, service delivery, and evidence-based advocacy for people with MSI in Malawi.

In view of the lack of accurate data on the prevalence and causes of MSI in Malawi, we conducted a survey of MSI using a reliable sampling methodology with a case definition and diagnostic criteria that could clearly be related to the classification system used in the ICF. The aim of this study was to assess to report the prevalence, impact, causes and service implications of MSI in Malawi. Data gathered will inform policy on advocacy and lobbying for appropriate resource allocation for MSI. To achieve this we chose to use a new survey tool developed in Rwanda by Atijosan et al. (2007).

## Methods

### Setting

Malawi has an estimated population of about 18.3 million (Nation estimates 2018 census). The country is divided in 3 administrative regions: The Northern, Central, and Southern Regions. The Central and Southern regions are the most densely populated with 6.4 and 6.8 million respectively [18]. Malawi has 28 districts and a total of 48,233 registered settlements. The vast majority of these are in the rural areas. About 90% of the population live in rural areas and are dependent mostly on subsistence farming [19].

## Sample selection

A sample size of 1,481 households was derived based on the following formula for calculation of household sample sizes:—$n_h$ = (84.5)(1-r)/(r)(p) [20] and assuming 95 percent level of confidence, a sample design effect of 2.0, a non-response multiplier of 1.1, an average household size of 6, and a margin of error of 10%. Based on estimates from Rwanda and Cameroon, r (a key indicator to be measured by the survey, being prevalence of musculoskeletal impairment for this study) is 5.4% [4] and since all the population will be targeted, p = 1. The formula therefore gives a sample size of n = (84.5)(1–0.054)/(0.054)(1.0) = 1,481 households.

We selected clusters across the whole country through probability proportional to size sampling with an urban/rural and demographic split that matched the national distribution of the population. Then individuals (both adults and children) were examined in their households by survey field teams.

The National Statistics Office provided a list of enumeration areas from the Malawi Census Board for 2008 national census records. These settlements were randomized through computer-generated random numbers, selecting 55 settlements as enumeration areas from each district in Malawi for this survey. Two or four households were randomly selected in each settlement depending on size. The randomization was based on a ground bottle spin and selecting the third or fifth house in the direction of spin depending on the size of the settlement. Subsequently the bottle spinning was repeated after the household interview to select the next household in the new direction of the spin. The next thirds household was then picked if in a smaller settlement, or fifth household if in a larger settlement, then repeating the process again to select the next household. All household members present were screened. For the youngest (age below five) household members, the guardian of the child was interviewed. People were eligible for inclusion if they lived in the household at least three months of the year. All the individuals in the final household were interviewed, and the number of people needed to complete the survey in the settlement was randomly selected for inclusion (e.g. if the final household included six people but only two were required to complete the number for the settlement then two out of the six were randomly selected for inclusion). If an eligible participant was absent the survey team paid one more visit to the household to examine him/her before leaving the area. If not found, information about his/her presumed MSI status was collected from relatives present.

## Musculoskeletal impairment assessment

The survey tool developed in Rwanda by Atijosan et al. (2007) fulfilled the proposed criteria and aims, and was therefore chosen for this study [21]. This screening tool was developed by orthopaedic surgeons together with physiotherapists and has been shown to have 99% sensitivity and 97% specificity with inter-observer Kappa scores of 0.90 for the diagnostic group. The team of data collectors screened all participants for MSI by asking them seven questions about difficulties using their musculoskeletal system and how long they had had these symptoms. Participants who answered "yes" to any of the questions were classified as cases, provided that the condition had lasted for more than one month or was considered permanent (Table 1). The questionnaire and other Rapid Assessment of Musculoskeletal Impairment (RAM 1& RAM 2) were installed on 17 tablet computers (iPad 2, Apple Inc.), using File Maker Pro 12.0v3 (File maker Inc., USA) software for data collection in English (see Appendix of S1 and S2 Files).

The fieldworkers visited households door-to-door and conducted the MSI screening in the household. The survey team was assisted in the village by a village guide, appointed by the

Table 1. Screening questionnaire.

| Screening for musculoskeletal impairment | Yes | No |
|---|---|---|
| 1. Is any part of your body missing or misshapen? | ❍ | ❍ |
| 2. Do you have any difficulty using your arms? | ❍ | ❍ |
| 3. Do you have any difficulty using your legs? | ❍ | ❍ |
| 4. Do you have any difficulty using any other part of your body? | ❍ | ❍ |
| 5. Do you need a mobility aid or prosthesis? | ❍ | ❍ |
| 6. Do you have convulsions, involuntary movement, rigidity or loss of consciousness? | ❍ | ❍ |
| If any of the answers are "yes": | | |
| 7. Has it lasted more than one month or is it permanent? | ❍ | ❍ |

village leaders. The purpose of the study and the examination procedure were explained to the subjects and verbal consent was obtained before examination.

## Screening for musculoskeletal impairment

**Standardized interview and examination protocol.** All cases were examined in more detail by the students using a standardized interview and physical examination protocol. Whenever in doubt, the students consulted a supervisor physically or by phone (calls or pictures). Only those who were able to respond to all the five dimensions of quality of life (EQ-5D-3L) were eligible.

The standardized examination protocol assessed the area affected, duration, etiology, diagnosis, severity, quality of life of the participants and treatment received and needed. The elements included in the interview and examination protocol are presented in Table 2

Quality of life: The EQ-5D-3L is a public domain quality of life questionnaire from the Euro-Qol group, which has been validated in a number of countries and cultural settings [24]. It allows the participant to indicate their health state by indicating the most applicable statement in five parameters, including mobility, self-care, usual activities, pain/discomfort, anxiety/depression, with a maximum score of 100 (best quality of life) and minimum score of 0

Table 2. Standardized interview and examination protocol.

| Elements | Definition |
|---|---|
| Physical assessment | Performance of physical tasks that require use of the musculoskeletal system, both lower and upper limb motor skills. (i.e. walking, standing, sitting, running etc.) |
| Anatomical location | Information of the affected part of the body (e.g. leg) and the nature of the problem (e.g. tumour) |
| Duration | The duration of the MSI, classified into a long (> 1 month) or short (<1 month) standing history |
| Etiology | Initiation and cause of the impairment (infection, violence etc.) |
| Diagnosis | Diagnosis categorized as: neurological, traumatic, congenital, metabolic, infective, or acquired non-traumatic non-infective. Within these categories an algorithm was created and used to give a specific diagnosis. Up to two diagnoses were permissible per case [21]. |
| Severity | Severity was classified as "mild", "moderate" or "severe according to ICF "[22]. |
| Quality of life (EQ-5D) | The Malawian version of the EQ-5D-3L questionnaire [23]. |
| Treatment received | Any known treatment given to the participant (medical or others) was recorded |
| Treatment needed | Treatment required by the participants was assessed according to Malawi standard treatment guideline |
| Barriers to treatment | Participants were asked one question about why they had not received treatment for their MSI. All responses (up to four options) were recorded on pre-coded forms |

(death). Severity was determined using the parameters for the percentage of function outlined in the WHO reference book International Classification of Functioning (ICF) [2]. A loss of function of 5–24% was mild, 25–49% was moderate and 50–90% was severe. The Malawian version of the EQ-5D-3L questionnaire was used [23]

### Data collection

Data collection was done by 32 third-year medical students. They all underwent a 14 days training supervised by two orthopaedic surgeons and two senior orthopaedic clinical officers on how to assess persons with musculoskeletal impairment and the use of the questionnaire and computer tablet. A pilot study/training was carried out in rural areas of the capital city, Lilongwe. The aim was to assess the examination process, function of the computer tablets and procedures. A second round of training was carried out as a refresher after the pilot study in preparation for the national survey.

Data were collected from 1st July to 30th August 2016. In some areas, local translators were hired to secure good communication between the interviewer and the household member. Each data collector covered approximately two households per day (10–12 participants), therefore 30–34 households were interviewed every day. Interviews took place in the interviewees' private homes. Data was checked and exported into the Excel (Microsoft 2010) pooled database at the end of each day, for data security and to assure the quality of the data collection [18]. A survey record was filled in for each eligible person that included: Demographic information (all participants), screening examination for MSI, a standardized interview and examination protocol for MSI, history of MSI (if not examined).

### Statistical analysis

Extrapolation was done based on study size compared to the population of Malawi. Chi-square test was used to test categorical variables. Odds ratio (OR) was calculated with a linear regression model adjusted for age, gender, location and education. EQ-5D index scores were calculated using the values from Zimbabwe [25, 26], as there are no values for Malawi, and Zimbabwe was considered the closest country. The statistical analyses were performed using IMB-SPSS Statistics, version 24.0 for Windows (IBM Corp, Armonk, NY, USA) and the statistical package R, version 3.4.0 (http://www.R-project.org). P-values less than 0.05 were considered statistically significant.

### Ethical approval

The approval to conduct this survey was granted by the College of Medicine Research and Ethics Committee (COMREC) and The Regional Committee for Medical and Health Research Ethics (REC Western Norway) in Norway. Consent to survey the districts and clusters were granted respectively by the District Commissioner and village head for each visited district and cluster.

Consent was obtained from the participants after explaining to them the goals and possible benefits of the study. Both verbal and written consent were obtained from adults (18 years of age and above), and assent were obtained from parents/guardians of children less than 18 years of age.

Data collectors were allowed to take photographs for teaching and discussion purposes after a verbal consent was granted from the participant. All those with manageable MSI were referred either to the MACOHA (Malawi Council of Handicapped) field workers (in the central region) or to district hospitals in the northern and southern region of Malawi for

appropriate action such as Physiotherapy, prosthetic and orthotic devices, mobility aids and orthopaedic surgery.

This study was funded by Norad through the Norhed programme.

## Results

The total number of included households was 1,481, with a total of 8,886 persons enumerated (with an average household size of six). 85 participants were excluded due to missing data. Among the 8,801 persons properly enumerated, 16 participants were not able to communicate (adequately), 64 refused to participate, and 173 were absent. Finally, 8,548 persons were screened or examined (response rate of 97.1%). The response rate was similar in women (97.5%) and men (96.7%). Among the participants that were enumerated, but not examined, eight (3.2%) were believed to have MSI. The age and gender distribution of the sampled population was similar to that of the national population (Table 3). During the national population and housing census enumeration process, the enumerators estimate the age of persons with unknown age based on past events or events of national interest (Nation estimates 2008 census).

### Prevalence of MSI

Of the 8,548 participants that were screened, 810 cases of MSI were diagnosed. This gave an overall prevalence of MSI of 9.5% (CI 8.9–10.1) (Table 2). The prevalence of MSI was higher among participants aged between 31 and 60 years (OR = 1.9, 1.5–2.5) and those over 60 years (OR = 5.7, 4.2–7.7) compared to the three youngest groups together (Fig 1). The prevalence of MSI was similar in men (9.3%) and women (9.6%). Persons without formal education were more likely to have an MSI (13.3%) compared to those with formal education levels (Table 4). The odds ratios were derived from logistic regression analyses (adjusted for age group, gender, location and education level).

### Prevalence of MSI by severity, and quality of life

MSI had an impact on the patients' quality of life. Patients with severe MSI had lower quality of life compared to patients with mild MSI (Table 5). Table 6 shows that all 5 dimensions of the EQ-5D were influenced by the degree of MSI. Some 25–30% of patients with severe MSI were

**Table 3. Age and gender composition of national and screened sample population.**

| Age groups | Male | | | Female | | | Total | | |
|---|---|---|---|---|---|---|---|---|---|
| | National | Enumerated Sample (%) | Screened Sample (%) | National | Enumerated Sample (%) | Screened Sample (%) | National | Enumerated Sample (%) | Screened Sample (%) |
| **0–10** | 3,282,887 | 1,163 (26.7%) | 1,132 (26.8%) | 3,197,698 | 1,050 (23.7%) | 1,033 (23.9%) | 6,480,585 | 2,213 (25.1%) | 2,165 (25.3%) |
| **11–20** | 1,992,015 | 1,265 (29%) | 1,217 (28.8%) | 2,054,034 | 1,179 (26.6%) | 1,131 (26.1%) | 4,046,049 | 2,444 (27.8%) | 2,348 (27.5%) |
| **21–30** | 1,380,453 | 690 (15.8%) | 660 (15.6%) | 1,452,729 | 772 (17.4%) | 755 (17.4%) | 2,833,182 | 1,462 (16.6%) | 1,415 (16.6%) |
| **31–40** | 928,658 | 451 (10.3%) | 441 (10.5%) | 1,002,444 | 535 (12.1%) | 526 (12.2%) | 1,931,102 | 986 (11.2%) | 967 (11.3%) |
| **41–50** | 587,303 | 337 (7.7%) | 330 (7.8%) | 635,670 | 333 (7.5%) | 325 (7.5%) | 1,222,973 | 670 (7.6%) | 655 (7.7%) |
| **51–60** | 332,188 | 188 (4.3%) | 182 (4.3%) | 365,001 | 243 (5.5%) | 237 (5.5%) | 697,189 | 431 (4.9%) | 419 (4.9%) |
| **>60** | 326,567 | 239 (5.5%) | 233 (5.5%) | 393,988 | 317 (7.1%) | 312 (7.2%) | 720,555 | 556 (6.3%) | 545 (6.4%) |
| **Unknown*** | | 29 (0.7%) | 24 (0.6%) | | 10 (0.2%) | 10 (0.2%) | | 39 (0.4%) | 34 (0.4%) |
| **Total** | 8,830,071 | 4,362 (100.0) | 4,219 (100.0) | 9,101,564 | 4,439 (100.0) | 4,329 (100.0) | 17,931,635 | 8,801 (100.0) | 8,548 (100.0) |

*participants with unknown age.

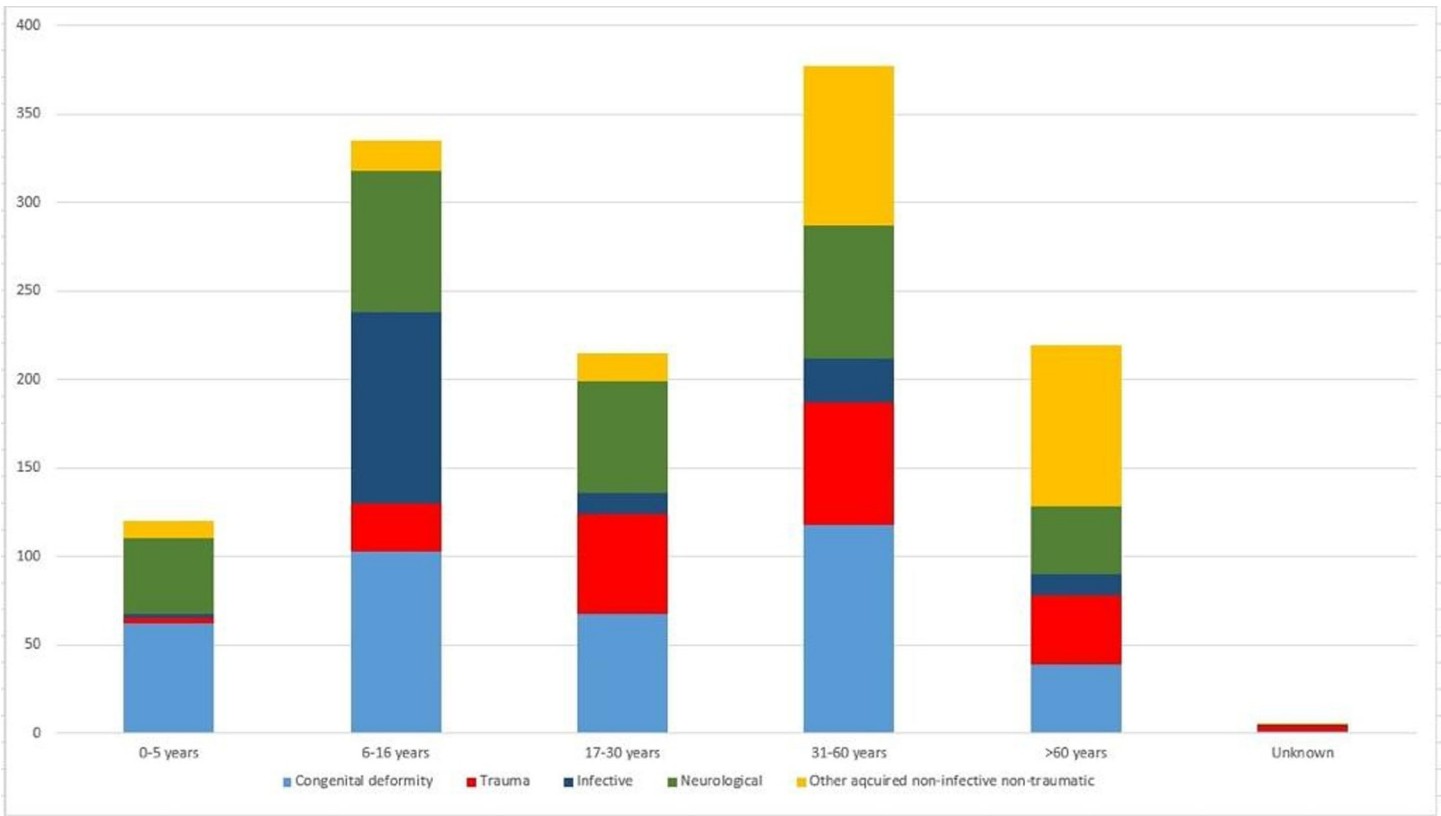

**Fig 1. Number and diagnostic categories of MSI, by age group.** Y-axis: Number of cases.

confined to bed, unable to wash or undress or unable to perform usual activities (Table 6). Further, a large proportion of patients with severe MSI had pain or anxiety/depression.

## MSI diagnoses

There were a total of 1,174 diagnoses for 810 individuals with MSI diagnosed (Table 7). Overall, 33.2% of MSIs were due to congenital causes, 25.6% were neurological in origin, 19.2% were acquired non-traumatic non-infective causes, 16.8% were due to trauma, and 5.2% due to infection. Congenital and neurological diagnoses remained relatively constant in all age groups. Acquired non-traumatic non-infective diagnoses were common in participants 31 years of age and above. However, trauma diagnoses were uncommon in participants 5 years of age and below.

## Treatment needed

In total, 503 treatments were needed for the 1,174 diagnoses (Table 8). The most common treatments needed were prosthetic and orthotic devices (33.1%), medication (26.6%), physical therapy (15%) and surgery (4%). Extrapolating these estimates to the entire population of Malawi, approximately 1,054,000 treatments are required, including 350,000 prosthetic and orthotic devices; 281,000 courses of medicine; 159,000 courses of physical therapy and 42,000 operations.

## Discussion

The main findings of this study are the high estimated prevalence of MSI in Malawi of 9.5% (CI 8.9–10.1) and the need for over one million interventions including medication, physical

**Table 4. Prevalence of MSI by age, gender, location and educational level of head of household.**

| Categories | | Total no Screened | No of MSI cases | Prevalence of MSI (95% CI) | Age and gender adjusted Odds Ratios (95% CI) |
|---|---|---|---|---|---|
| | Total | 8,548 | 810 | 9.5 (8.9–10.1) | |
| Age groups, years | 0–5 | 1,109 | 76 | 6.9 (5.4–8.3) | 1.0 (0.8–1.3) |
| | 6–16 | 2,539 | 160 | 6.3 (5.4–7.2) | 0.9 (0.7–1.2) |
| | 17–30 | 2,280 | 154 | 6.8 (5.2–7.8) | 1 |
| | 31–60 | 2,041 | 254 | 12.4 (11.0–13.9) | 2.0 (1.6–2.4) |
| | >60 | 545 | 161 | 29.5 (25.7–33.4) | 5.8 (4.5–7.4) |
| | Unknown* | 34 | 5 | 14.7 (2.8–26.6) | 2.4 (0.9–6.2) |
| Gender | Male | 4,219 | 393 | 9.3 (8.4–10.2) | 1 |
| | Female | 4,329 | 417 | 9.6 (8.8–10.5) | 1.0 (0.8–1.1) |
| Location | Rural | 8,058 | 773 | 9.6 (8.9–10.2) | 1 |
| | Urban | 415 | 33 | 8.0 (5.3–10.6) | 0.9 (0.6–1.2) |
| | Mobile, urban/ rural | 75 | 4 | 5.3 (0.2–10.4) | 0.5 (0.2–1.4) |
| Education level of head of household** | No formal education | 2,074 | 276 | 13.3 (11.8–14.8) | 1 |
| | Primary school | 5,025 | 449 | 8.9 (8.1–9.7) | 0.7 (0.6–0.8) |
| | Secondary school | 1,249 | 74 | 5.9 (4.6–7.2) | 0.4 (0.3–0.6) |
| | University / college | 98 | 7 | 7.1 (2.0–12.2) | 0.5 (0.2–1.0) |
| | Unknown | 102 | 4 | 3.9 (0.2–7.2) | 0.3 (0.1–0.8) |

* Participants with unknown age.

** The education number of the head of household accounted for each of the screened participants.

therapy, prosthetic orthotic devices and surgery to alleviate the burden. Most cases of MSI were moderate (54%) or severe (28%) according to the ICF classification. These MSIs greatly affect people's quality of life, having impact on all five dimensions of EQ-5D. The severity and scale of the burden of MSI in Malawi is likely to affect society at large [27, 28] and have a negative impact on the development of the communities and of Malawi as a country [27].

The factors that were associated with increased risk of MSI were increased age and lack of formal education. In the former, this was the result of an increase in acquired non-traumatic non-infective degenerative conditions. These results were in line with the findings reported in a similar study in Rwanda [8]. Another study on disability transitions and health expectancies among adults 45 years and older in Malawi has shown that the risks of experiencing functional limitations due to poor physical health are high in this population, and the onset of physical disabilities happens early in life [29]. Lack of education is likely to coincide with farming as an occupation, rural location, hard work and poor ergonomics that could to lead to MSI. There was a tendency towards more MSI in rural areas. But the low number of cases from urban

**Table 5. Impact of MSI on quality of life.**

| MSI status | Number | Mean EQ-5D index score | Std Error of the mean | 95% CI |
|---|---|---|---|---|
| Mild MSI | 108 | 81.6 | 1.63 | 78.4–84.8 |
| Moderate MSI | 329 | 69.4 | 0.98 | 67.4–71.3 |
| Severe MSI | 167 | 49.2 | 2.15 | 45.0–53.4 |
| Total | 604* | 66.0 | 0.96 | 64.1–67.9 |

*Out of 810 cases of MSI, 604 participants were able to respond to all the five dimensions of the EQ-5D-3L.

**Table 6. Distribution of patients in each level of the 5 dimensions of EQ-5D-3L according to MSI severity level.**

| EQ-5D | Mild MSI | Moderate MSI | Severe MSI | p-value* |
|---|---|---|---|---|
| **Mobility** | | | | <0.001 |
| • **No problems in walking about** | 82 (64.1%) | 117 (32.9%) | 31 (16.6%) | |
| • **Some problems in walking about** | 44 (34.4%) | 224 (62.9%) | 110 (58.8%) | |
| • **Confined to bed** | 2 (1.6%) | 15 (4.2%) | 46 (24.6%) | |
| **Self-care** | | | | <0.001 |
| • **No problems with self-care** | 104 (82.5%) | 211 (59.1%) | 56 (31.1%) | |
| • **Some problems with self-care** | 19 (15.1%) | 124 (34.7%) | 77 (42.8%) | |
| • **Unable to wash or dress** | 3 (2.4%) | 22 (6.2%) | 47 (26.1%) | |
| **Usual activities** | | | | <0.001 |
| • **No problem in performing usual activities** | 72 (63.7%) | 105 (30.5%) | 21 (12.1%) | |
| • **Some problem in performing usual activities** | 40 (35.4%) | 216 (62.8%) | 98 (56.6%) | |
| • **Unable to perform usual activities** | 1 (0.9%) | 23 (6.7%) | 54 (31.2%) | |
| **Pain/Discomfort** | | | | <0.001 |
| • **No pain or discomfort** | 64 (53.8%) | 126 (35.6%) | 63 (34.4%) | |
| • **Some pain or discomfort** | 51 (42.9%) | 206 (58.2%) | 81 (44.3%) | |
| • **Extreme pain or discomfort** | 4 (3.4%) | 22 (6.2%) | 39 (21.3%) | |
| **Anxiety/Depression** | | | | <0.001 |
| • **Not anxious or depressed** | 82 (70.1%) | 139 (40.3%) | 45 (25.3%) | |
| • **Moderately anxious or depressed** | 33 (28.2%) | 185 (53.6%) | 81 (45.5%) | |
| • **Extremely anxious or depressed** | 2 (1.7%) | 21 (6.1%) | 52 (29.2%) | |

* p-values were calculated using the Chi square.

areas could be due to the lower number of people living in the urban areas. However the prevalence of MSI was similar in men and women.

With regard to causes of MSI, congenital and neurological causes were the most common diagnostic categories in all age groups, followed by acquired non-infective non-traumatic causes, especially in the middle aged and elderly population. In the latter the most common individual diagnosis was joint problems (9% of MSI diagnoses). The Global Burden of Disease Study 2015 estimated that the most important contributors to global years lived with disability were musculoskeletal disorders (18.5%) [30]. Neck and Lower back pain were estimated in 2013 as the leading cause of years lived with disability in Cameroon [31]. The prevalence of MSI was shown to be increasing with age in our study. This finding supports studies of older person's health in Botswana and Malawi that showed an increased probability of musculoskeletal disease and functional limitations [29, 32]. As the prevalence of musculoskeletal disorders increases with age, there will be a significant increase in requirements for health care and community support in the future.

Musculoskeletal disease is known to be a major cause of morbidity and mortality, especially in LICs with non-traumatic musculoskeletal disease estimated to account for 6.8% of all Disability-Adjusted Life Years (DALY) lost [1]. The overall prevalence of MSI in this study is almost double the 5.2% reported in Rwanda [8], but similar to the 11.6% reported in Fundong District, North-West Cameroon which used the same survey methods. The proportion of severe MSI was much higher in this study compared to what has been reported in Cameroon (2.4%) and Rwanda (8.4%). The reasons for this are unclear, but may, in part, reflect the long distances patients need to walk to seek medical attention in Malawi [33], and also the lack of medical expertise and equipment in the district hospitals and in the country overall. A study conducted by The College of Surgeons of East, Central, and Southern Africa (COSECSA) in 267 hospitals in east central and southern Africa has shown that current capacity to treat trauma and orthopaedic conditions is very limited, with particular areas of concern being manpower, training, facilities, and equipment [34]. However, the assessment of severe MSI deserves further attention in future studies.

**Table 7. Cause of MSI in survey, and extrapolated to population of Malawi.**

| Diagnosis | Number | Total in category (%) | Extrapolated number of that diagnosis in Malawi to nearest 1000 |
|---|---|---|---|
| **Congenital deformity** | | **390 (33.2%)** | 818,000 |
| Syndactyly | 43 | | |
| Polydactyly | 74 | | |
| Other Upper Limb deformity | 32 | | |
| Club foot | 41 | | |
| Other Lower Limb deformity | 56 | | |
| Spine deformity | 125 | | |
| Other congenital deformity | 19 | | |
| **Trauma** | | **198 (16.8%)** | 415,000 |
| Burn contracture | 24 | | |
| Fracture non-/ malunion | 48 | | |
| Spine injury | 1 | | |
| Head injury | 6 | | |
| Tendon/nerve injury | 45 | | |
| Amputation | 46 | | |
| Joint chronic dislocation | 21 | | |
| Other chronic joint injury | 7 | | |
| **Neurological** | | **299 (25.6%)** | 627,000 |
| Epilepsy | 106 | | |
| Polio (sequelae) | 33 | | |
| Para/quadra/Hemiplegia | 61 | | |
| Cerebral palsy | 65 | | |
| Peripheral nerve palsy | 12 | | |
| Other neurological MSI | 22 | | |
| **Infective** | | **62 (5.2%)** | 130,000 |
| Bone infection limb | 19 | | |
| Joint infection | 12 | | |
| Spine infection | 16 | | |
| Soft tissue infection | 15 | | |
| **Other acquired non-infective non- traumatic** | | **225 (19.2%)** | 472,000 |
| Angular limb deformity | 24 | | |
| Degenerative and other Joint problem | 108 | | |
| Spine pain | 3 | | |
| Skin/ soft tissue/ bone swelling | 19 | | |
| Limb swelling | 57 | | |
| Limb pain | 6 | | |
| Other acquired spine deformity | 8 | | |
| **Total** | **1174** | | |

Prosthetic and orthotic devices (P&O), physical therapy (rehabilitation), mobility aids, medication and surgery were the most frequently recommended treatments for the people identified with MSI in this study. These data have shown a significant treatment need for MSI in Malawi. With estimated 42,000 surgical operations needed, and only 11 orthopaedic surgeons in the country, it is obvious that Malawi is in dire need of scaling up surgical services rapidly and this concurred with the COSECSA study [34]. Some participants were very sick and there was no appropriate treatment to offer in our setting. However, counseling was provided to them and their relatives.

**Table 8. Treatment needed among cases with MSI in survey and extrapolated to population of Malawi.**

| Treatment modality | Number of cases in survey needing that treatment modality | Extrapolated number in country needing that treatment modality (based on 2016 population estimates) |
|---|---|---|
| Medication | 134 | 281,000 |
| Physiotherapy | 76 | 159,000 |
| Appliance | 36 | 75,000 |
| Prosthesis | 72 | 151,000 |
| Orthosis (splints/ braces) | 95 | 199,000 |
| Surgery | 20 | 42,000 |
| Wheelchair /Tricycle | 39 | 82,000 |
| Permanent care | 6 | 3,000 |
| None | 25 | 52,000 |
| Total | 503 | |

The burden of MSI is predicted to increase as the population of Malawi, and the World, is aging. Musculoskeletal impairment or disability related to trauma are also rapidly increasing in the future due to the rise in Road Traffic Injuries in our country [35] and worldwide [30]. Therefore, there is a need to recognize musculoskeletal conditions as a national and global public health priority. Solutions to fill this health service gap are needed. With the prevalence of MSI being higher among people living in rural areas, access to health services may be encouraged through health programs and support in rural communities. A wide range of ergonomically designed tools could be made available to ease agricultural work for those in need of this. Developing programs that serve populations at the district level [36], where needs can be assessed, and resources identified, may improve access to preventive services and rehabilitation, and facilitate transfer to tertiary hospitals when needed. Continued support of task shifting through the orthopaedic clinical officer program [37] at the district level is a natural part of this until a sufficient number of surgeons have been trained. However, to rapidly scale up the surgical specialist service in a severely resource limited country like Malawi, specialist services need to be concentrated to a few training centres while these grow into sustainably sized units. These units can scale up production more rapidly at lower investment costs and provide short stay trauma and orthopaedic services that serve the districts until enough surgeons can be trained also for the district hospitals.

This population-based survey used a standardized examination protocol to provide estimates of musculoskeletal impairment in the country. The data from this study provides important information to assist planning of P&O services, provision of mobility aids, rehabilitation, medical, and surgical services for persons with MSI in Malawi. The need for medical services such as surgery, drug supply, and rehabilitation has been be estimated, and the more detailed need for equipment and other assistive devices (e.g. appliances, orthoses, prostheses and wheelchairs) can be estimated from this information.

This study did have some limitations on the probability proportional to size sampling; diagnostic tools were limited to history and clinical examination, which restricted the identification of conditions that need complex investigations and data on other socioeconomic factors like occupational status/ type of occupation were not collected. Due to the long distances in some areas, the call back at a few households where people were unavailable was not achieved. Our demographic data were very limited. However, this study was a nationwide survey with a representative sample of people of all ages who were enumerated and examined. The response rate was high, and the sample was representative of the national population for both age and

gender. This has reduced the likelihood of selection bias. The outcome definition was undertaken by well-trained medical students, using an examination protocol and screening tool, which was used in a similar study in Rwanda [21]. The inter-observer agreement between the data collectors was high as all were closely monitored and supervised.

## Conclusion

This study has uncovered a high prevalence of MSI in Malawi and contributed data to the epidemiology of MSI nationally and globally. The Quality of Life of those with severe MSI was considerably affected. Increasing age and lack of formal education were factors that were associated with an increased risk of having MSI. The huge burden of musculoskeletal impairment in Malawi is mostly unattended, revealing an urgent need to scale up orthotics & prosthetics, physical & occupational therapy and surgical services in the country.

## Supporting information

**S1 File. Rapid assessment of Musculoskeletal impairment.**
(PDF)

**S2 File. Rapid assessment of Musculoskeletal impairment.**
(PDF)

## Author Contributions

**Conceptualization:** Leonard Banza Ngoie, Geir Hallan, Sven Young.

**Data curation:** Leonard Banza Ngoie, Geir Hallan, Jan-Erik Gjertsen, Nyengo Mkandawire, Sven Young.

**Formal analysis:** Eva Dybvik, Jan-Erik Gjertsen.

**Investigation:** Leonard Banza Ngoie.

**Project administration:** Leonard Banza Ngoie.

**Writing – review & editing:** Leonard Banza Ngoie, Eva Dybvik, Geir Hallan, Jan-Erik Gjertsen, Nyengo Mkandawire, Carlos Varela, Sven Young.

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
