## [Decision Letter · Decision Letter 0]

10 May 2020

PONE-D-20-05125

Prevalence, Causes and Impact of Musculoskeletal Impairment in Malawi: A National Cluster Randomized Survey

PLOS ONE

Dear Dr Ngoie,

Thank you for submitting your manuscript to PLOS ONE. After careful consideration, we feel that it has merit but does not fully meet PLOS ONE’s publication criteria as it currently stands. Therefore, we invite you to submit a revised version of the manuscript that addresses the points raised during the review process.

We would appreciate receiving your revised manuscript by Jun 24 2020 11:59PM. To enhance the reproducibility of your results, we recommend that if applicable you deposit your laboratory protocols in protocols.io, where a protocol can be assigned its own identifier (DOI) such that it can be cited independently in the future. For instructions see: http://journals.plos.org/plosone/s/submission-guidelines#loc-laboratory-protocols

We look forward to receiving your revised manuscript.

Kind regards,

Subas Neupane

Academic Editor

PLOS ONE

Additional Editor Comments:

I agree with the reviewers suggestions to improve the quality of the manuscript. Please answer each of the reviews's questions carefully and revise the manuscript accordingly. I have here listed some additional comments.

Although the study title claimed that it is cluster randomized trial, but the methodology needs a lot of improvements to reflect the study design. The analysis should also take into account the study design it means that the data must be weighted.

The aim of the study is not very clear both in the abstract and in the main texts.

Introduction should present well the rationale of the study.

Methods need some more information on how the outcome was defined and measured, briefly about the main statistical technique used for the analysis etc. There are many errors and typos in the results, which should be corrected.

Explain, why only very limited demographic data are presented. Did author measure only these sets of variables? As such it is very limited and many details are lacking.

Statistical analysis part needs some more information as also pointed out by reviewer 2.

2. We noticed you have some minor occurrence of overlapping text with the following previous publication, which needs to be addressed:

https://www.ncbi.nlm.nih.gov/pmc/articles/PMC2483936/?amp=&tool=pubmed

In your revision ensure you cite all your sources (including your own works), and quote or rephrase any duplicated text outside the methods section. Further consideration is dependent on these concerns being addressed.

3. Please provide additional details regarding participant consent. In the ethics statement in the Methods and online submission information, please ensure that you have specified how verbal consent was documented and witnessed). As your study included minors, please state whether you obtained consent from parents or guardians.

4. Please refer to any post-hoc corrections to correct for multiple comparisons during your statistical analyses. If these were not performed please justify the reasons. Please refer to our statistical reporting guidelines for assistance (https://journals.plos.org/plosone/s/submission-guidelines.#loc-statistical-reporting).

5. Please amend your authorship list in your manuscript file to include author Leonard Banza Ngoie.

6. Please ensure that you refer to Figure 1 in your text as, if accepted, production will need this reference to link the reader to the figure.

Reviewers' comments:

Reviewer's Responses to Questions

**Comments to the Author**

1. Is the manuscript technically sound, and do the data support the conclusions?

Reviewer #1: Yes

Reviewer #2: Yes

2. Has the statistical analysis been performed appropriately and rigorously? 

Reviewer #1: Yes

Reviewer #2: Yes

3. Have the authors made all data underlying the findings in their manuscript fully available?

Reviewer #1: Yes

Reviewer #2: Yes

4. Is the manuscript presented in an intelligible fashion and written in standard English?

Reviewer #1: Yes

Reviewer #2: Yes

5. Review Comments to the Author

Reviewer #1: There are too few studies on the burden of diseases from low income countries. World vide there is especially a lack of studies concerning musculoskeletal disorders. This manuscript is very interesting and very well written on this important topic. The study design is proper and the high response rate is remarkable (97%) and ending in a high number of participants (8548) in all ages. This is partly due to a relevant sampling method from survey field teams going from household to household. The method part is very well described in the text. The results are interesting and mostly clearly written, including both tables and a figure. One question is why do you come up with MSI according to ICF only as in the first sentence of the discussion? The ICF classification is important and could be showed and explained, more than just in appendices, in both methods and as a part of the results. The discussion is mostly on care and rehabilitation but could also include plausable causes for the MSI and aspects of prevention. E.g. most of the participants are living in rural areas and I suppose they work mostly with agricultural tasks. These are mostly one important cause to musculoskeletal problems world vide and one important method of prevention is ergonomics (physical). Training of physicians and other health care workers in simply ergonomics may be an important part of prevention, but this is, of course, not the main aim of this study.

Reviewer #2: Manuscript PONE-D-20-05125 entitled “Prevalence, Causes and Impact of Musculoskeletal Impairment in Malawi: A National Cluster Randomized Survey”

The work has covered an important issue of musculoskeletal impairment in one of the African country where this kind of research are limited. This survey report nicely reflect the musculoskeletal health of the people there. I would like to suggest the authors to address the changes according to the comments listed below:-

Overall, the manuscript has several qualities; however, there are several areas to be improved in the manuscript. The most important concern from my side is on methods part: some information is missing (example explanation of variables).

Abstract: The abstract looks good, has explained, and expressed much. In result ….. the

prevalence of MSI increased with age, and was similar in men (9·3%) and women (96%), check the %. Further, in conclusion some statements used are very irrelevant (second line)

Introduction: Introduction is wisely written. However, the authors shall be consistent in using the terms, for instance, musculoskeletal impairments have been studied and they have introduced the terms musculoskeletal diseases quite often.

*More focus should be given to impairments rather than disease or disorder.

*Second paragraph does not fit here, which, could have been more wisely used to connect with the outcome of the study.

*Or it would be better to state that musculoskeletal impairment includes disease, disorder, disability…..

*The authors could more explicitly state early on what were the major results/findings of several surveys of physical disability (as they have stated few were done) that were done before in Malawi.

*In the present form, the introduction is difficult to follow and missing some connection, please revise it thoroughly.

Methods: Though author have tried best to explain methods precisely. Explanation of variable and their construction are missing in some instances….

*Please explain more about EQ-5D-3L in this section, referring is not enough. Likewise, description of mild, moderate and severe MSI is missing.

*In case of sample size calculation (please give the values as *n= (84·5) (1- 0·054)/(0·054) (1.0) and could you please recheck that you have used a sample design effect of 2·0…

* Some form of cluster sampling used in the sample design of household surveys will help in reduction of cost and besides that, there are many other pros of cluster sampling in itself. However, the use of probability proportional to size sampling to select the clusters could permit the sampler to exercise greater control over the ultimate sample size or overall sample size. In addition, it could decrease the reliability of the sample because people living in the same cluster could be homogeneous or have the same background/ characteristics (clustering effect) and mostly this effect is balanced or compensated in the sample design by increasing the sample size accordingly. Did the authors experience this situation and applied some solution? if not it could be briefly noted on limitations part.

*Sub heading Screening for musculoskeletal impairment: …. “ This screening tool was developed by orthopaedic surgeons together with physiotherapists and has been shown to have 99% sensitivity and 97% specificity with interobserver Kappa scores of 0·90 for the diagnostic group……” It is quite surprising for me, how an interview/ a self-reported response on seven questions had such a profound level of sensitivity and specificity…. did author check for some bias, if there were any?

*Please rewrite the statistical analysis part, open up about the approaches used. Please ELABORATE

Results: Result section is easy to follow.

* Was the data on other socioeconomic factors like occupational status/ type of occupation collected? It would be more relevant to see the results (if available)

*Avoid the use of p-values in table 3, 95% CI is sufficient

* In table 3, age and sex adjusted estimates (ORs and 95% CIs) is confusing, is it so that OR and 95% CI for age was adjusted by gender, location, and educational level; for gender was adjusted by age, location and edu level…and so on?

* Use gender instead of sex, if it is reported as “gender” on first column

*Check the misprinting in table 4

*Again I would suggest that the explanation of EQ-5D index score calculation and extrapolation is important. It will help readers (easy to follow the results).

Discussion: There discussion has a logical presentation of results and comparison in the global and local scenario.

* The authors switch between the discussions of findings. Please discuss more from the point of age, gender and educational differences.

* “The burden of MSI is predicted to increase as the population of Malawi, and the World, is aging. Musculoskeletal impairment or disability related to trauma are also rapidly increasing in the future due to the rise in Road Traffic Injuries in our country31 and worldwide.23,27 Therefore, there is a need to recognize musculoskeletal conditions as a national and global public health priority. Solutions to fill this health service gap are needed. With the prevalence of MSI being higher among people living in rural areas, access to health services may be encouraged through health programs and support in rural communities…….” ..Here two very different scenarios are described together, …please mention these two circumstances separately ….

* “This population-based survey used a standardized examination protocol to provide estimates of musculoskeletal impairment in the country. The data from this study provide important information to assist planning of P&O devices, mobility aids, rehabilitation, medical, and surgical services for persons with MSI in Malawi. For example, the need for medical services such as surgery, drug supply, and rehabilitation can be estimated, and the need for equipment and other assistive devices (e.g. appliances, orthoses, prostheses and wheelchairs) can also be estimated from this information. Therefore, production and supply of these items can be anticipated. Medical services can be used to measure the capacity of existing services in the country, for advocacy and planning of future service provision such as training of both paramedical and medical personnel (e.g. surgeons, prosthetists & orthotists, physical therapists), building of new health care facilities or improving the existing ones to treat the burden of MSI.”………… For me this is little ambitious. In my opinion, this is certainly a broad piece of work that is carried out with precision plus methodological qualities; however, there are still some gaps in the survey that could be fulfilled.

*Please separately specify the strengths and weakness of the study.

6. PLOS authors have the option to publish the peer review history of their article (what does this mean?). If published, this will include your full peer review and any attached files.

Reviewer #1: No

Reviewer #2: No

---

## [Author Response · Author response to Decision Letter 0]

20 Oct 2020

Dear Editor, 

Thank you for the opportunity to resubmit this article after revisions, and for extending the deadline due to the Covid-19 situation in Malawi and elsewhere. The authors apologise for the very late resubmission. This is due to the dire situation over the last few months in Malawi after Covid-19 hit the country. The situation is slowly recovering but the backlog of surgical patients is huge. 

We do believe that this study has uncovered some dramatic information about the burden of disabilities in Malawi that really needs to be communicated, and hope you will now find the manuscript worthy of publication in PLOS ONE. We thank you and the reviewers for your time and efforts to provide helpful comments, and have tried to address all these below and through the revisions in the manuscript:

Although the study title claimed that it is cluster randomized trial, but the methodology needs a lot of improvements to reflect the study design. The analysis should also take into account the study design it means that the data must be weighted.

Thank you. We agree that analysis should be weighted, but data on population according to the clusters in Malawi was not possible to obtain.

The aim of the study is not very clear both in the abstract and in the main texts.

We agree this might have been presented more clearly and have edited the text. We hope it is now clearer.

Introduction should present well the rationale of the study.

Thank you for pointing this out. We have revised the introduction and hope it is clearer now.

Methods need some more information on how the outcome was defined and measured, briefly about the main statistical technique used for the analysis etc. There are many errors and typos in the results, which should be corrected.

We have in the revised manuscript made changes in the text to clarify this, and errors have been corrected to our best ability. We hope that the presentation now is acceptable to your journal.

Explain, why only very limited demographic data are presented. Did author measure only these sets of variables? As such it is very limited and many details are lacking.

The data that is presented was the only demographic data measured. The data included age, gender, location of settlement and educational level. Similar studies also have had a limited set of demographic data. A survey of this scale and design is a huge logistic exercise anywhere, but in a resource limited environment like Malawi even more so, and there needs to be a balance between the desire for comprehensive information and time and funding limitations. We believe the presented demographics include the essentials, but of course, a more comprehensive set of data would have been better. The scarcity of details has now been acknowledged as a study limitation in the appropriate section of our manuscript.

Statistical analysis part needs some more information as also pointed out by reviewer 2.

Thank you. This section has been revisited with the second author (ED) who is a bio statistician with the Norwegian Arthroplasty Register. 

3. Please provide additional details regarding participant consent. In the ethics statement in the Methods and online submission information, please ensure that you have specified how verbal consent was documented and witnessed). As your study included minors, please state whether you obtained consent from parents or guardians.

Consent forms were used. Both verbal and written Consent was obtained from adults (18 years of age and above), and assent was obtained from parents/guardians of children less than 18 years of age. This has now been described in the ethical approval section.

4. Please refer to any post-hoc corrections to correct for multiple comparisons during your statistical analyses. If these were not performed please justify the reasons. Please refer to our statistical reporting guidelines for assistance (https://journals.plos.org/plosone/s/submission-guidelines.#loc-statistical-reporting).

Since no multiple comparisons were done, post hoc correction was not relevant here.

5. Please amend your authorship list in your manuscript file to include author Leonard Banza Ngoie.

Thank you for noticing this accidental omission. The author list now includes Leonard Banza Ngoie

6. Please ensure that you refer to Figure 1 in your text as, if accepted, production will need this reference to link the reader to the figure.

Reference to Figure 1 has been included.

Changes have been made (see Methods and Appendix sections)

Reviewer #1: 

One question is why do you come up with MSI according to ICF only as in the first sentence of the discussion? 

The ICF classification is important and could be showed and explained, more than just in appendices, in both methods and as a part of the results. 

We agree that the ICF classification should have been included earlier in the manuscript. In the revised manuscript the ICF classification has been explained in the Methods.

The discussion is mostly on care and rehabilitation but could also include plausible causes for the MSI and aspects of prevention. E.g. most of the participants are living in rural areas and I suppose they work mostly with agricultural tasks. These are mostly one important cause to musculoskeletal problems world vide and one important method of prevention is ergonomics (physical). Training of physicians and other health care workers in simply ergonomics may be an important part of prevention, but this is, of course, not the main aim of this study.

This is a good point. We have added a comment on this in the text.

(see paragraph 6 of the discussion section)

Reviewer #2: 

Overall, the manuscript has several qualities; however, there are several areas to be improved in the manuscript. The most important concern from my side is on methods part: some information is missing (example explanation of variables).

Thank you for pointing this out. We agree, and the methods section has been revised.

Abstract: The abstract looks good, has explained, and expressed much. In result ….. the

prevalence of MSI increased with age, and was similar in men (9·3%) and women (96%), check the %.

Thank you for noticing this. Checked and corrected.

Further, in conclusion some statements used are very irrelevant (second line)

We agree. These irrelevant statements have been removed.

Introduction: Introduction is wisely written. However, the authors shall be consistent in using the terms, for instance, musculoskeletal impairments have been studied and they have introduced the terms musculoskeletal diseases quite often.

*More focus should be given to impairments rather than disease or disorder.

*Or it would be better to state that musculoskeletal impairment includes disease, disorder, disability…..

We agree that there was some inconsistency here and have tried to remedy this in the revised manuscript.

*Second paragraph does not fit here, which, could have been more wisely used to connect with the outcome of the study.

Another good point. Changes have been made. 

*The authors could more explicitly state early on what were the major results/findings of several surveys of physical disability (as they have stated few were done) that were done before in Malawi.

We have attempted to clarify this in the introduction, although most of these studies were based on physical disability in general.

*In the present form, the introduction is difficult to follow and missing some connection, please revise it thoroughly.

The introduction has been revised. We hope it is clearer now.

Methods: Though author have tried best to explain methods precisely. Explanation of variable and their construction are missing in some instances….

*Please explain more about EQ-5D-3L in this section, referring is not enough. Likewise, description of mild, moderate and severe MSI is missing.

Changes have been made. A new section on EQ-5D-3L has been inserted in Methods. Also, a definition of mild, moderate and severe MSI has been added. 

*In case of sample size calculation (please give the values as *n= (84·5) (1- 0·054)/(0·054) (1.0) and could you please recheck that you have used a sample design effect of 2·0…

When focusing on household survey sample size, in terms of households it was calculated using the formula: nh = (z 2 ) (r) (1-r) (f) (k)/ (p) (n) (e 2 )

The reference quoted recommends parameters as follows: z-statistics should be 1.96 for 95-percent level of confidence, as default value of f (sample design effect) should be set at 2.0 since there is no empirical data from previous or related surveys, k is multiplier to account for anticipated rate of non-responders, a value of 1.1 would be a conservative choice for a undeveloped country as Malawi. Average household size is given by n, and margin of error to be attained is denoted by e, which is recommended to set as 10 percent of r, e=0.10r.

nh = (3.84) (1-r) (1.2) (1.1)/ (r) (p) (6) (.01)

reduced to 

nh = (84.5) (1-r)/ (r) (p)

where r is an estimate of a key indicator in the survey and p is proportion of the total population accounted for by the target population.

nh = (84.5) (1-0.054)/ (0.054) (1.0) = 1,481

* Some form of cluster sampling used in the sample design of household surveys will help in reduction of cost and besides that, there are many other pros of cluster sampling in itself. However, the use of probability proportional to size sampling to select the clusters could permit the sampler to exercise greater control over the ultimate sample size or overall sample size. In addition, it could decrease the reliability of the sample because people living in the same cluster could be homogeneous or have the same background/ characteristics (clustering effect) and mostly this effect is balanced or compensated in the sample design by increasing the sample size accordingly. Did the authors experience this situation and applied some solution? if not it could be briefly noted on limitations part.

Thank you. Limitation of probability proportional to size sampling has now been mentioned in the discussion (last paragraph). 

*Sub heading Screening for musculoskeletal impairment: …. “ This screening tool was developed by orthopaedic surgeons together with physiotherapists and has been shown to have 99% sensitivity and 97% specificity with interobserver Kappa scores of 0·90 for the diagnostic group……” It is quite surprising for me, how an interview/ a self-reported response on seven questions had such a profound level of sensitivity and specificity…. did author check for some bias, if there were any?

This statement is referenced in the manuscript and refers to the paper by Atijosan et al (2007) where the tool was validated. The tool was developed and published together with recognised scholars from the University of Oxford and London School of Hygiene and Tropical Medicine, and we have accepted the findings as presented in their work. 

*Please rewrite the statistical analysis part, open up about the approaches used. Please ELABORATE 

The statistics section has been revised with our bio statistician co-author (ED).

Results: Result section is easy to follow.

* Was the data on other socioeconomic factors like occupational status/ type of occupation collected? It would be more relevant to see the results (if available)

This was not the main aim of this study. We agree that would be relevant but unfortunately this information was not available.

*Avoid the use of p-values in table 3, 95% CI is sufficient

Corrected. P-values have been removed from table 3 (now table 4)

* In table 3, age and sex adjusted estimates (ORs and 95% CIs) is confusing, is it so that OR and 95% CI for age was adjusted by gender, location, and educational level; for gender was adjusted by age, location and edu level…and so on?

We used logistic regression analyses meaning that the OR is adjusted for in the way you describe above. This has been described in the Statistics paragraph and in the table text for Table 3 (now table 4). 

* Use gender instead of sex, if it is reported as “gender” on first column

Corrected

*Check the misprinting in table 4

Corrected (now table 5)

*Again I would suggest that the explanation of EQ-5D index score calculation and extrapolation is important. It will help readers (easy to follow the results).

Changes have been made (“screening for musculoskeletal impairment” section).

Discussion: There discussion has a logical presentation of results and comparison in the global and local scenario.

* The authors switch between the discussions of findings. Please discuss more from the point of age, gender and educational differences.

We have made some changes in the discussion section to accommodate this point. Please see paragraph 2.

* “The burden of MSI is predicted to increase as the population of Malawi, and the World, is aging. Musculoskeletal impairment or disability related to trauma are also rapidly increasing in the future due to the rise in Road Traffic Injuries in our country31 and worldwide.23,27 Therefore, there is a need to recognize musculoskeletal conditions as a national and global public health priority. Solutions to fill this health service gap are needed. With the prevalence of MSI being higher among people living in rural areas, access to health services may be encouraged through health programs and support in rural communities…….” ..Here two very different scenarios are described together, …please mention these two circumstances separately ….

Thanks for the observation. We have dealt with the two circumstances separately in the discussion section (paragraph 6).

* “This population-based survey used a standardized examination protocol to provide estimates of musculoskeletal impairment in the country. The data from this study provide important information to assist planning of P&O devices, mobility aids, rehabilitation, medical, and surgical services for persons with MSI in Malawi. For example, the need for medical services such as surgery, drug supply, and rehabilitation can be estimated, and the need for equipment and other assistive devices (e.g. appliances, orthoses, prostheses and wheelchairs) can also be estimated from this information. Therefore, production and supply of these items can be anticipated. Medical services can be used to measure the capacity of existing services in the country, for advocacy and planning of future service provision such as training of both paramedical and medical personnel (e.g. surgeons, prosthetists & orthotists, physical therapists), building of new health care facilities or improving the existing ones to treat the burden of MSI.”………… For me this is little ambitious. In my opinion, this is certainly a broad piece of work that is carried out with precision plus methodological qualities; however, there are still some gaps in the survey that could be fulfilled.

We do agree with this observation, however, this is what we were able to achieve at our level best. Some of the mentioned gaps are among the study’s weaknesses. Despite this, we do believe this study brings forward very important information for policy makers in Malawi, and regionally and that some of the gaps are areas for potential future research.

*Please separately specify the strengths and weakness of the study.

 We have made the suggested changes in the revised manuscript.

Thank you again for the opportunity to resubmit this manuscript and many thanks to the reviewers for their helpful feedback. 

Best regards

Leonard Banza

---

## [Decision Letter · Decision Letter 1]

10 Nov 2020

PONE-D-20-05125R1

Prevalence, Causes and Impact of Musculoskeletal Impairment in Malawi: A National Cluster Randomized Survey

PLOS ONE

Dear Dr. Ngoie,

Thank you for submitting your manuscript to PLOS ONE. After careful consideration, we feel that it has merit but does not fully meet PLOS ONE’s publication criteria as it currently stands. Therefore, we invite you to submit a revised version of the manuscript that addresses the points raised during the review process.

ACADEMIC EDITOR: In abstract, the methods part, please explain briefly how the study subjects were randomized and how musculoskeletal impairments were measured in ICF criteria and the main statistical methods used to analyze the data.

We look forward to receiving your revised manuscript.

Kind regards,

Subas Neupane

Academic Editor

PLOS ONE

Additional Editor Comments (if provided):

Thank you for the revised manuscript. Both the reviewer are happy with the revision the authors have made. I have now only few minor issues in the manuscript.

Reviewers' comments:

Reviewer's Responses to Questions

**Comments to the Author**

1. If the authors have adequately addressed your comments raised in a previous round of review and you feel that this manuscript is now acceptable for publication, you may indicate that here to bypass the “Comments to the Author” section, enter your conflict of interest statement in the “Confidential to Editor” section, and submit your "Accept" recommendation.

Reviewer #1: All comments have been addressed

Reviewer #2: All comments have been addressed

2. Is the manuscript technically sound, and do the data support the conclusions?

Reviewer #1: Yes

Reviewer #2: Yes

3. Has the statistical analysis been performed appropriately and rigorously? 

Reviewer #1: Yes

Reviewer #2: Yes

4. Have the authors made all data underlying the findings in their manuscript fully available?

Reviewer #1: Yes

Reviewer #2: Yes

5. Is the manuscript presented in an intelligible fashion and written in standard English?

Reviewer #1: Yes

Reviewer #2: Yes

6. Review Comments to the Author

Reviewer #1: The authors has noted my comments in the methods and discussion and changed the text accordingly to an acceptable extent

Reviewer #2: (No Response)

7. PLOS authors have the option to publish the peer review history of their article (what does this mean?). If published, this will include your full peer review and any attached files.

Reviewer #1: No

Reviewer #2: No

---

## [Author Response · Author response to Decision Letter 1]

19 Nov 2020

Dear Editor, 

Thank you for the opportunity to resubmit this article after revisions, following a brief feedback from you.

 We thank you and the reviewers for your time and efforts to provide helpful comments, and have tried to address all these below and through the revisions in the manuscript:

ACADEMIC EDITOR: In abstract, the methods part, please explain briefly how the study subjects were randomized and how musculoskeletal impairments were measured in ICF criteria and the main statistical methods used to analyse the data.

The abstract has been completely revised and most of the points raised were addressed.

Thank you again for the opportunity to resubmit this manuscript and many thanks to the reviewers for their positive response to our previous manuscript. 

Best regards

Leonard Banza

---

## [Editor Report · Decision Letter 2]

24 Nov 2020

Prevalence, Causes and Impact of Musculoskeletal Impairment in Malawi: A National Cluster Randomized Survey

PONE-D-20-05125R2

Dear Dr. Ngoie,

We’re pleased to inform you that your manuscript has been judged scientifically suitable for publication and will be formally accepted for publication once it meets all outstanding technical requirements.

Kind regards,

Subas Neupane

Academic Editor

PLOS ONE
---

## [Editor Report · Acceptance letter]

2 Dec 2020

PONE-D-20-05125R2 

Prevalence, Causes and Impact of Musculoskeletal Impairment in Malawi: A National Cluster Randomized Survey 

Dear Dr. Ngoie:

I'm pleased to inform you that your manuscript has been deemed suitable for publication in PLOS ONE. Congratulations! Your manuscript is now with our production department. 

Kind regards, 

on behalf of

Dr. Subas Neupane 

Guest Editor

PLOS ONE